# Exploring the Effects of Tourism Development on Air Pollution: Evidence from the Panel Smooth Transition Regression Model

**DOI:** 10.3390/ijerph19148442

**Published:** 2022-07-11

**Authors:** Jun Zhang, Youhai Lu

**Affiliations:** 1School of Tourism and Urban-Rural Planning, Zhejiang Gongshang University, Hangzhou 310018, China; 2School of Geography and Ocean Science, Nanjing University, Nanjing 210023, China; dg1827022@smail.nju.edu.cn

**Keywords:** tourism development, PM_2.5_ emissions, PSTR model, tourism-induced EKC hypothesis, China

## Abstract

Based on the theoretical framework of the Environmental Kuznets Curve (EKC), this study investigates whether tourism development can decrease air pollution. This study applies the panel smooth transition regression approach and panel data for 2005–2019 from 283 prefecture-level cities in China to examine the nonlinear effect of tourism development on PM_2.5_, emissions. Our results reveal that the effects of tourism on PM_2.5_, emissions vary according to the modes of tourist arrivals. At the national level, the effect of tourism on PM_2.5_ emissions exhibits an inverted-U shape. At the regional level, tourism exerts a U-shaped impact on PM_2.5_ emissions in eastern China, and tourism is nonlinearly negatively associated with PM_2.5_ emissions in central and western China. An important theoretical contribution of our study is the proposal and validation of the U-shaped tourism-induced EKC hypothesis.

## 1. Introduction

In recent years, the tourism industry has been regarded as one of the strategic pillar industries of the Chinese national economy, partially because it exerts an increasingly important impetus to promote economic growth and increase employment [1,2]. According to data from of the Ministry of Culture and Tourism of the People’s Republic of China, the number of domestic tourist arrivals reached 6.0 billion in 2019, an increase of 8.4%, and the total tourism revenue reached 961.3 billion USD, an annual increase of 11%. The tourism industry makes a comprehensive contribution of 1.59 trillion USD to the GDP, accounting for 11.05% of the GDP. There are 79.87 million direct and indirect employees in the tourism industry, accounting for 10.31% of the total workforce in China.

Another reason for the importance of the tourism industry is that tourism was once characterized as a ‘smokeless industry’. Tourism-induced air pollution (including NOx, PM_2.5_, PM_10_, CO, and SO_2_ emissions) is marginal in comparison with that of the manufacturing industry [3,4]. Therefore, tourism is a promising economic and environmental alternative to Fordist-style resource extraction and production [5]. Many tourists consider air quality when deciding on their destinations [6]. Therefore, clean air at a destination can improve a tourist’s experience. In contrast, travelling in a polluted environment depresses travelling experiences and negatively affects tourists’ willingness to revisit [7]. This creates a necessity for clean tourist destinations to cater to tourist demand. Thus, tourism contributes to mitigating air pollution. Moreover, the air quality of tourism-oriented cities is generally superior to those of manufacture-oriented cities.

However, the rapid development of the tourism industry has resulted in controversies regarding tourism being environmentally friendly. Tourism-induced pollution has become a prevalent topic in academic tourism circles. Some scholars believe that the current economic growth and development driven by tourism comes at the cost of pollution and environmental degradation [8,9]. Tourism-induced air pollution emissions are greater than those of other service sectors. This may be because tourism is an energy-intensive industry [9], and almost all sectors (e.g., catering, transportation, retailing, and accommodation) within the tourism industry have experienced increased demands for heavy energy, thereby resulting in large air pollution emissions [10]. A study found that almost 70% of the carbon emissions generated by the tourism industry originates from the combustion of fuels used for accommodation, transportation, and land use [11,12]. Higham et al. (2016) found that the demand for transportation, catering, accommodation, and other tourism processes significantly contributed to global air pollution emissions [13]. Therefore, excessive and unduly expanding tourism pose severe challenges to the environmental quality of tourist destinations.

These conflicting views from existing studies may lead to inconsistent results regarding whether and to what extent tourism significantly affects air pollution. Although the relationship between tourism and air pollution has been well-documented, there are still drawbacks. First, previous studies have mainly focused on the influence of air pollution on tourism, and less attention has been devoted to the effects of tourism on air pollution. Second, regarding the proxy variables of air pollution, most studies focus on CO_2_ emissions, neglecting other air pollutants such as PM_2.5_, PM_10_, and SO_2_. Finally, a large body of empirical studies have already proven that tourism development will lead to severe externalities that will threaten air quality [14,15], but this influence may not be simply linear [16,17]. The hypothesis of the tourism-induced environmental Kuznets curve (EKC), which depicts the inverted U-shaped relationship between tourism and air pollution [12], is widely applied to test nonlinear impacts. However, whether the tourism-induced EKC hypothesis also holds for other pollutants, such as PM_2.5_, has rarely been investigated.

Our study empirically investigates the effect of tourism development on air pollution based on panel data from 283 cities in mainland China between 2005 and 2019. PM_2.5_ emissions are indicators of air pollution because PM_2.5_, the culprit of air pollution in Chinese cities [8], can exert severe harm to human health and the environment more than other air pollutants. According to the WHO (2016), high PM_2.5_ concentrations can induce strokes, lung cancer, heart disease, chronic obstructive pulmonary disease, pneumonia, and other diseases. A study that went beyond examining the nonlinear tourism-air pollution nexus by introducing quadratic terms into linear models [8,18] found that an endogeneity problem, such as ordinary least squares, fixed effects, and panel data regression models, may exist in linear models and can lead to biased and inconsistent results [19,20]. The frontier econometric technique of the panel smooth transition regression (PSTR) model is employed. To achieve the research objective, we attempt to answer the following questions using the empirical results: Does tourism development exert an impact on PM_2.5_ emissions under different levels of tourism specialization? Does the tourism-induced EKC hypothesis hold true for different regions? If this is the case, what is the underlying mechanism? What are the characteristics of the spatial distribution of this influence? These answers will facilitate a profound understanding of the relationship between tourism and air pollution, which will contribute to ensuring the healthy development of the tourism industry in the foreseeable future.

To the best of our knowledge, this study is novel for reasons as follows: First, regarding the tourism-induced EKC hypothesis, this research provides new insights regarding the influence of tourism development on air pollutant PM_2.5_, thereby validating the tourism-induced EKC hypothesis for China and further proposing an extended tourism-induced EKC model. Second, instead of traditional econometric methods, this study applies the PSTR model to explore the nonlinear impact of tourism development on air pollution under the different threshold values of tourism specialization. In the PSTR model, the relationship between air pollution and tourism development smoothly transitions from a high tourism development regime to a low tourism development regime. Additionally, the PSTR model can effectively address the possible problem of endogeneity among variables [21]. Finally, because of the internally heterogeneous characteristics across regions in China’s vast territory, it is necessary to implement a case-by-case analysis on a regional scale [22]. This study divides China’s 283 cities into 3 distinct regions, including 100 east cities, 107 central cities, and 76 west cities, as these 3 separate groups have different levels of air pollution and tourism development. To reveal these disparities, it is important to explore the nonlinear effects of tourism development on air pollution within a more homogenous group of cities (i.e., with similar conditions). 

The remainder of this paper Is organized as follows. Section 2 reviews the effect of tourism on air pollution and the tourism-induced EKC hypothesis. Section 3 describes the theoretical analysis, methods, and data. Section 4 presents the empirical results. The discussion and conclusions are presented in Section 5.

## 2. Literature Review

### 2.1. The Effect of Tourism on Air Pollution

Although most studies find evidence supporting a significant impact of tourism development on air pollution, the direction of its influence is still under discussion. Overall, these empirical results can be classified into three strands of research. The first strand suggests that tourism has a positive effect on air pollution. Many studies find that tourism leads to an increase in air pollution [16,23]. Koçak et al. (2020) and Katircioğlu (2014) determined that tourism increases energy consumption and CO_2_ emissions. Some studies using pollutants other than CO_2_ have found a positive effect of tourism on destination air pollution. For example, Saenz-de-Miera and Rosselló (2014) used a semi-parametric approach to model the impacts of air pollutant PM_10_ in Mallorca, Spain [14]. The results indicated that a 1% increase in tourist arrivals could be related to a 0.45% increase in PM_10_ emissions. Robaina et al. (2020) studied the influence of tourism on PM_10_ concentrations as a representative of air quality in five European countries. The empirical results prove that tourism growth can deteriorate air quality in Austria and Italy [24].

The second strand considers the negative impact of tourism on air pollution. Lee and Brahmasrene (2013) found that tourism was negatively correlated with CO_2_ emissions in 27 European countries. Specifically, a 1% increase in tourism resulted in a 0.11% decrease in CO_2_ emissions [25]. Bojanic and Warnic (2020) examined the impact of tourism on global greenhouse gas (GHG) emissions. Their findings indicate that countries with higher tourism densities have lower GHG emissions and higher environmental performances [26]. Tian et al. (2021) found that a 1% increase in tourism development leads to a 0.05% decrease in CO_2_ emissions in the long run in G20 economies [10]. Ciarlantini et al. (2022) found some differences in the impacts of international and local tourism on air pollution, that is, that international tourism negatively affects NO_x_, PM_10_, and PM_2.5_ emissions, while local tourism increases their emissions [27].

The third strand shows a nonlinear relationship between tourism and air pollution. Paramati et al. (2017) concluded that, when CO_2_ emissions exceed a certain threshold, the estimated coefficient of tourism on CO_2_ emissions begins to decrease owing to government policies, especially in developed countries [16]. Sherafatian-Jahromi et al. (2017) found an inverted U-shaped EKC between tourism and CO_2_ emissions in five Southeast Asian countries [17]. Balsalobre-Lorente et al. (2020) used OECD countries as a case study and found that tourism has an inverted U-shaped impact on CO_2_ emissions [28].

In China, only a few studies have considered the effect of tourism on air pollution, and the conclusions are inconsistent. For example, Bi and Zeng (2019) explored the effects of tourism on carbon emissions in China from nonlinear and spatial perspectives and found that a significant inverse U-shaped relationship exists between them. Moreover, they found that tourism has a spatial nonlinear spillover effect on carbon emissions [29]. Similar results obtained by Zeng et al. (2021) suggest that the direct effect of tourism development on PM_2.5_ emissions exhibits an inverted U-shaped curve [8].

### 2.2. Tourism-Induced EKC Hypothesis

The Environmental Kuznets Curve (EKC) hypothesis, proposed by Grossman and Krueger (1991), suggests that, when a country’s economic development is at a low level, environmental deterioration intensifies with economic growth; when economic development reaches a specific threshold, that is, after reaching a critical, or ‘inflection point’, environmental pollution continuously decreases with economic growth. This implies that environmental quality is gradually improving. It demonstrates an inverted U-shaped pattern between economic growth and pollution. Thereafter, a large number of studies have emerged to validate the EKC hypothesis, but the results have been inconsistent [30]. There are more than four types of EKCs in the literature, such as the inverted U-shaped, U-shaped, N-shaped, and M-shaped curves [31,32].

The tourism-induced EKC hypothesis, first proposed by Katircioglu (2014) [33], provides a theoretical basis for our study. Until the recent decade, this hypothesis has drawn some attention from scholars in the context of rapid tourism development [16,34,35]. It indicates that environment pollution follows an increasing and then decreasing trend with the development of the tourism industry. Subsequently, many studies have tested the tourism-induced EKC hypothesis in different countries and regions [8]. Akadiri et al. (2019) confirmed the globalization-tourism-induced EKC hypothesis and found that international tourism and the squared term of real income have an inverse significant impact on carbon emissions [36]. Katircioglu et al. (2018) investigated the role of tourism development on the ecological footprint quality of the top 10 tourist destinations and found that tourism development has an inverted U-shaped relationship with the ecological footprint, supporting the tourism-EKC hypothesis [8]. Gao and Zhang (2021) validated the tourism-induced EKC hypothesis for southern Mediterranean countries, but failed to support the hypothesis for the northern Mediterranean region [37]. Yildırım et al. (2021) tested the tourism-EKC hypothesis for 15 Mediterranean countries and found that tourist arrivals increased carbon emissions until a certain threshold was reached and then decreased carbon emissions above this level [38]. Ciarlantini et al. (2022) explored the relationship between air pollution and tourism growth in five European countries and failed to validate the tourism-EKC hypothesis for any of the countries [27].

## 3. Theoretical Analysis, Method, and Data

### 3.1. Theoretical Analysis

Various theories have been used to reveal the theoretical connections between tourism development and air pollution and to explain the shape of the EKC, such as the dynamic general equilibrium theoretical model [39] and Dutch disease theory [40]. There are mainly three shapes in tourism literature describing the relationship between tourism and pollution: two linear (positive and negative) relationships and an inverted U-shaped relationship (see Figure 1a–c). This study attempts to extend the conventional theoretical tourism-induced EKC framework, that is, the U-shaped tourism-induced EKC hypothesis, and then verifies its validity through the following empirical analysis.

In contrast to the conventional EKC hypothesis, the U-shaped tourism-induced-EKC hypothesis (see Figure 1d) suggests that air pollution continues to decrease as tourist arrivals increase in regime 1 until tourist arrivals reach a certain threshold. When tourist arrivals exceed the threshold (regime 2), the effect of tourist arrivals on air pollution becomes positive. This is because, when the number of tourists exceeds a destination’s carrying capacity and the government’s environmental governance capacity, tourism development will lead to severe air pollution. This study hypothesizes that, as tourist arrivals are associated with air pollution, once tourist arrivals reach a certain threshold, the air quality will decrease. Thus, a U–shaped relationship may exist between tourism development and air pollution, namely, the U–shaped tourism induced-EKC hypothesis.

### 3.2. Method 

#### 3.2.1. Panel Smooth Transition Regression Model 

To account for the potentially non-linear impact of tourism development on air pollution, this study adopts the panel smooth transition regression (PSTR) approach developed by Gonzalez et al. (2005) [41], which can resolve the heterogeneity problem of different units in a nonlinear framework [42]. In this study, the PSTR model had the following advantages: Firstly, the PSTR model allows for parameter heterogeneity in the panel model [21,43]. Second, the PSTR model allows for a smooth transition between regimes [44]. Finally, transitions within the PSTR model could effectively verify the tourism-EKC hypothesis that exhibits a U-shaped or inverted U-shaped pattern. Based on the above considerations, it is appropriate to implement a PSTR model to capture the nonlinear and regime-switching effects of tourism on air pollution. 

The general equation of the PSTR model with two more regimes can be defined as follows:(1)$$y_{it}=\mu_{i}+\varphi x_{it}+\overset{\gamma }{\underset{j=1}{\sum }}\beta_{j}x_{it}g_{j}\left(q_{it}{}^{\left(j\right)};\;\gamma_{j};c_{j}\right)+\alpha z_{it}+\varepsilon_{it},$$
where *i* = 1, …, *N* and *t* = 1, …, *T*; *N* and *T* represent the number of cities and period of the panel, respectively. In Equation (1), *y_it_* denotes the dependent variable and *x_it_* denotes a vector of the time-varying independent variable. φ and β denote the estimated coefficients of the linear and nonlinear components, respectively. $\mu_{i}\;$stands for the individual fixed effect, *z_it_* denotes the control variables and *α* is their estimated coefficients. $\varepsilon_{it}$ is the random errors. $g_{it}\left(q_{it}{}^{\left(j\right)};\;\gamma_{j};c_{j}\right)$ stands for the transition function of the transition variable $q_{it}$. Further, the equation of $g_{it}\left(q_{it}{}^{\left(j\right)};\;\gamma_{j};c_{j}\right)$ can be expressed as follows:(2)$$g_{it}\left(q_{it}{}^{\left(j\right)};\;\gamma_{j};c_{j}\right)=\left(1+\exp\left(-\gamma \overset{m}{\underset{j=1}{\prod }}\left(q_{it}{}^{\left(j\right)}-c_{j}\right)\right)\right),\;\gamma >0,c_{1}\leq c_{2}\leq c_{3}\ldots \leq c_{m}$$
where $\gamma_{j}\;$is the number of transition function which describes transition slope, $c_{j}$ stands for the location parameter and *m* is the number of $g_{it}\left(q_{it}{}^{\left(j\right)};\;\gamma_{j};c_{j}\right)$. Gonzalez et al. (2005) [41] found that it is sufficient to consider only the cases of *m* = 1 or *m* = 2 that capture the nonlinearities caused by regime switching. *m* = 1 corresponds to a logistic PSTR model, and *m* = 2 represents a logistic quadratic PSTR specification [45], where there is a U-shaped or inverted U-shaped pattern described by the shape of the transition function [46].

#### 3.2.2. Linearity Test and Non-Remaining Nonlinearity Test 

Before the estimation, according to Colletaz and Hurlin (2006) [47], a linearity test of the PSTR model must be conducted by applying the Lagrange multiplier (LM), F-version LM (LMF), and pseudo-LR tests to check whether the regime-switching effect is effective [48]. The equations of the LM, LMF, and LR tests can be specifically constructed as follows:(3)$$\mathrm{LM}=\frac{\mathrm{TN}\left({\mathrm{SSR}}_{0}-{\mathrm{SSR}}_{1}\right)}{{\mathrm{SSR}}_{0}},$$
(4)$${\mathrm{LM}}_{F}=\frac{\mathrm{TN}\left({\mathrm{SSR}}_{0}-{\mathrm{SSR}}_{1}\right)/k}{{\mathrm{SSR}}_{0}/\left(\mathrm{TN}-N-K\right)},$$
(5)$$\mathrm{LR}=-2\left[\log\left({\mathrm{SSR}}_{1}\right)-\log\left({\mathrm{SSR}}_{0}\right)\right],$$
where *k* denotes the number of independent variables, SSR0 denotes the sum of the squared residuals under H0 (linear panel model with individual effects), and SSR_1_ denotes the sum of the squared residuals under H1 (PSTR model with two regimes). Under the null hypothesis, the LM and LR statistics are χ^2^(*k*) distributions, whereas the LMF statistics are distributed with F values (*k*, TN-N-K). If the null hypothesis (H0) does not pass the significance test, then the PSTR model is linear. If the null hypothesis (H0) is rejected, the PSTR model is nonlinear. Next, the non-remaining nonlinearity test proposed by Fouquau et al. (2008) [49] can be used to determine the number of transition functions and the number of regimes to be included in the PSTR model [50].

### 3.3. Variable Selection and Data Sources

This study uses panel data from 283 prefecture-level cities in mainland China from 2005 to 2019. In the regression model, the annual average PM_2.5_ concentrations (in micrograms per cubic meter, μg/m^3^) are used to measure the level of air pollution because PM_2.5_ concentrations are widely monitored and their data are continuously collected by governments worldwide, making them easily obtainable. In current tourism and environmental research, it is a common practice to use PM_2.5_ concentrations as a proxy for air pollution [8,51]. Some scholars have found that tourism exerts a positive effect on PM_2.5_ emissions [51,52]. However, Zeng et al. (2021) found that the direct effect of tourism on PM_2.5_ emissions exhibits an inverted U shape [8]. The PM_2.5_ data for 2005–2019 were obtained from the measurement data of global surface PM_2.5_ concentration by atmospheric composition analysis group of Washington University (https://sites.wustl.edu/acag/datasets/surface-pm2-5/, accessed on 12 January 2022). Following the study of Xu et al. (2020) [51], this study employs ArcGIS software to vectorize these raster data into the annual average concentrations of PM_2.5_.

The core explanatory variable is the level of tourism development, which is represented by the ratio of total tourist arrivals to the local inhabitants (TA) [1]. The total tourist arrivals are comprised of local and international tourist arrivals. Annual data on total tourist arrivals for each city were derived from the China City Statistical Yearbook (2006–2020) and the Statistical Yearbook of each province (2006–2020).

Further, the economic development level (PGDP), population density (DENS), R&D investment (TECH), fixed capital investment (INVEST), transportation development (TRAFF), and green coverage (GREEN) have significant impacts on air pollution [8,53,54,55]. Transportation development is proxied by the ratio of the total passenger traffic volume of highways, railway transport, and civil aviation to local inhabitants. Therefore, these variables set as the control variables are included in the PSTR model. The data for these control variables were obtained from the China City Statistical Yearbook (2006–2020). To eliminate possible heteroskedasticity in the data, all variables were logarithm-sized. Table 1 presents the descriptive statistics of the study variables.

### 3.4. Model Setting

To explore the linear and nonlinear effects of tourism on air pollution, we followed the model of Stochastic impacts by regression on population, affluence, and technology (STIRPAT model) developed by York et al. (2003) [56] to specify the basic and logarithmic model as follows:(6)$$I_{i,t}=\alpha P_{i,t}^{a}A_{i,t}^{b}T_{i,t}^{c}\mu_{i,t},$$
(7)$$LnI_{i,t}=\alpha +aLnP_{i,t}+bLnA_{i,t}+cLnT_{i,t}+\mu_{i,t},$$
where *P*, *A*, and *T* represent the population, affluence, and technology, respectively; *a*, *b*, and *c* are their estimated coefficients; *α* denotes the constant; *μ* is the error term; and *I* and *t* denote the city and year, respectively. According to Liddle (2014) [57], Ahmad and Ma (2021) [12], and Zeng et al. (2021) [8], tourist arrivals as indicators of affluence and other control variables are included in the STIRPAT model. The linear model can be rewritten as follows:(8)$$LnPM=\alpha +aLnDENS_{i,t}+bLnTA_{i,t}+cLnTECH_{i,t}+dLnPGDP_{i,t}+eLnINVEST_{i,t}+fLnTRAFF_{i,t}+\;gLnGREEN_{i,t}+\mu_{i,t}.$$

In the PSTR model, it can further be specified as follow: (9)$$LnPM=\alpha +a_{0}LnDENS_{i,t}+b_{0}LnTA_{i,t}+c_{0}LnTECH_{i,t}+d_{0}LnPGDP_{i,t}+e_{0}LnINVEST_{i,t}+f_{0}LnTRAFF_{i,t}+\;g_{0}LnGREEN_{i,t}+\sum_{j=1}^{\gamma }\left[a_{1}LnDENS_{i,t}+b_{1}LnTA_{i,t}+c_{1}LnTECH_{i,t}+d_{1}LnPGDP_{i,t}+e_{1}LnINVEST_{i,t}+\;f_{1}LnTRAFF_{i,t}+g_{1}LnGREEN_{i,t}\right]{*\;g}_{j}\left(q_{it}{}^{\left(j\right)};\;\gamma_{j};c_{j}\right)+\mu_{i,t}$$

## 4. Empirical Results

Figure 2 shows the spatial distribution of PM_2.5_ concentrations and total tourist arrivals in 2005 and 2019. A darker color in the figures implies higher PM_2.5_ concentration and total tourist arrival values. China has suffered severe PM_2.5_ air pollution in the past 15 years, demonstrating a deterioration of air quality. Highly polluted cities are located in the North China Plain and Sichuan Basin. There was a slight change in the spatial distribution of PM_2.5_ concentrations among cities between 2005 and 2019. The annual average PM_2.5_ concentration for the entire area decreased from 45.79 μg/m^3^ in 2005 to 33.79 μg/m^3^. Nevertheless, it can still harm human health because the annual average is above 10 μg/m^3^ [51].

From 2005 to 2019, the tourism industry at the city level developed rapidly. According to the official statistics of China, the total number of total tourist arrivals increased from 1.2 billion in 2005 to 6 billion in 2019 (more than a 500% increase). There were significant differences in the levels of tourist arrivals among the cities. The degree of high tourist arrivals increased throughout the period, depicting a hierarchical diffusion trend from east to west. Due to unique resource endowments, location conditions, and transport accessibility, the high-value areas of total tourist flows are scattered in Beijing–Tianjin, the Yangtze River Delta, Pearl River Delta urban agglomerations [58], and the Chengdu-Chongqing Economic Circle. As suggested by Xu et al. (2020) [51] and Yang and Wong (2012) [58], changes in tourist arrivals exhibit a core-periphery evolutionary pattern that centers on these core cities and diffuses to the surrounding cities.

As a prerequisite for empirical analysis, it is necessary to test the order of integration of these variables to avoid a spurious regression. This study utilizes four unit-root tests, including the LLC, IPS, Fisher-ADF, and Fisher-PP tests, to check sequence stationarity in their levels and first differences. Table 2 presents the results of the stationarity tests. Most variables (lnTA, lnPGDP, lnINVEST, lnTECH, and lnGREEN) pass the stationary test both in their levels and first differences, indicating that the variables are stable. Three variables of lnPM, lnDENS, and lnTRAFF failed to reject the null hypothesis of stationarity in their levels, but pass the stationary test at a 1% level after the first-order difference, indicating that they are I (1). 

Furthermore, the panel cointegration test was used to verify whether a long-term equilibrium relationship exists among the studied variables [59]. The panel Kao cointegration test in Table 2 passed the cointegration test at a 1% statistical level, confirming that there is a long-term stable link among the variables. Therefore, the PSTR analysis is appropriate. 

After performing the stationarity test, it is essential to test whether the empirical analysis based on the PSTR model is appropriate using linearity and remaining nonlinearity tests. This study used lnTA as the transition variable to investigate the effect of tourism development on air pollution under different levels of tourism arrivals. Table 3 shows the results of the linearity and remaining nonlinearity tests. The null hypothesis of linearity (H0: r = 0 vs. H1: r = 1) was rejected for the LM, LMF, and LRT tests, indicating that the effect of tourism development on air pollution is nonlinear. Thus, this study determined the number of transition functions. The results in Column 2 show that the null hypothesis of the non-remaining nonlinearity test (H0: r = 1 vs. H1: r = 2) cannot be rejected for the LMF test, indicating that the PSTR model has only one transition function. Finally, AIC and BIC were used to select the number of location parameters. According to Table 3, the values of AIC and BIC at *m* = 1 are −4.668 and −4.644, respectively, which are less than their values of −4.667 and −4.642 under *m* = 2, respectively. This implies that the optimum number of location parameters is 1. Therefore, the PSTR model with one transition function (r = 1) and one transition location (*m* = 1) was preferred and selected. 

Column 3 in Table 4 reports the non-linear effects of tourism on PM_2.5_ emissions, estimated using the panel threshold regression (PTR) model. The threshold variable TA has a single threshold of 2.045 and produces two regimes for the PTR model. When the TA is less than 2.045, the estimated coefficient of lnTA is −0.067 and is significant at a 1% statistic level, implying that a 1% increase in PM_2.5_ emissions may significantly lead to a 0.067% decrease in the number of tourist arrivals. When the TA exceeds 2.045, the coefficient is negative and statistically significant, with an effect size of −0.97, which is stronger than that in the first regime. The control variables lnPGDP, lnTECH, lnINVEST, and lnTRAFF have positive effects on the lnPM, showing that economic development, R&D investment, capital investment, and transportation can significantly increase PM_2.5_ emissions.

Columns 4 and 5 show the estimated results of the PSTR model with one transition function and one transition location. The location parameter c was 2.294, indicating that the PSTR model transitioned to a double-regime mode. The transition parameter γ is 0.419, indicating that the transition between the two regimes is smooth. 

In terms of the effect of tourism development on PM_2.5_ emissions, when lnTA is less than 2.294 (TA < 9.915), the estimated coefficient of lnTA is 0.098 and is significant at 1% level, indicating that, under the low-tourism development regime, a 1% increase in the ratio of tourist arrivals to local inhabitants can induce a 0.098% increase in PM_2.5_ emissions. A possible reason is that, in cities with low levels of urban tourism development, many short-sighted behaviors, such as over-exploitation and disorderly competition in the tourism industry, a large number of high-energy consumption and high-pollution enterprises, and imperfect environmental regulations, lead to higher PM_2.5_ emissions [8]. When lnTA exceeds 2.294 (TA > 9.915), the estimated coefficient of lnTA is −0.155, which is significant at a 1% statistical level. This result indicates that, under the high-tourism development regime, tourism development can exert a significantly negative effect on PM_2.5_ emissions. Therefore, there is an inverted U-shaped link between tourism and PM_2.5_ emissions, verifying the presence of an inverted U-shaped EKC hypothesis for tourism [8,60]. For each city, the estimated coefficients of the lnTA were further interpreted as individual elasticities (Figure 3a). We find that the individual elasticities of the lnTA decrease and change from positive to negative with an increase in the lnTA. Figure 3b shows the spatial distribution of high-to-low tourism regimes based on the average TA in each city, showing that 224 cities are in the low-tourism development regime and 60 cities are in the high-tourism development regime. 

Among all the control variables, this study finds that the economic development (lnPGDP) has a positive significant effect on PM_2.5_ emissions under the low-tourism development regime, but its impact is significantly negative under the high-tourism development regime. One possible reason for this is that tourism-oriented economic development can decrease air pollution in local cities. Variable transportation development (lnTRAFF) is positively correlated with PM_2.5_ emissions under both regimes, but its coefficient becomes nonsignificant under the high tourism development regime. 

One may be concerned about the existence of a heterogeneous relationship between tourism and air pollution when considering different regions as destinations. Therefore, this study further divided the entire sample into the three sub-samples of easter, central, and west cities and examines heterogeneity across different regions. The main estimation results for the East, Central, and West cities based on the PSTR model are presented in Table 5. 

For the East cities of China, there are two location parameters, 2.362 and 2.796, which transition to a triple-regime PSTR model. Interestingly, this indicates that tourism-induced air pollution is U-shaped (see Figure 4a), which contradicts the conclusion based on the full sample. Specifically, when the threshold variable lnTA is less than 2.362, the estimated coefficient of lnTA is equal to −0.357 and is significant at a 1% statistical level, indicating that, under the low-tourism development regime, tourism development can reduce PM_2.5_ emissions. When the lnTA is between 2.362 and 2.796, the impact is still negative but becomes less sensitive (−0.028). When the lnTA is higher than 2.796, the coefficient is significant (0.344) at a 1% level, indicating that, under the high-tourism development regime, tourism can increase PM2.5 emissions. These empirical results support the U-shaped tourism-induced EKC hypothesis. According to the sample of the east cities, most cities with a high regime (lnTA > 2.796), including Beijing, Benxi, Huzhou, Sanya, Xiamen, and Zhoushan, also have high PM_2.5_ emissions.

For the Central and West cities, both PSTR models have double regimes. In the model for the Central cities, the effects of the lnTA are negative under both regimes, but the impact becomes weaker and nonsignificant under the high tourism development regime (Figure 4b). In the model for China’s West cities, the effects of the lnTA are significantly negative under both regimes, and their impact becomes stronger with an increase in the lnTA (Figure 4c).

A robustness check must be conducted to ascertain the reliability of the estimated results. We used the contribution of the total tourism revenue to the GDP (lnTR) as an important indicator of tourism development and as a substitute for the lnTA. Table 6 presents the robustness results of the effect of tourism revenue on PM_2.5_ emissions in China. The direction and magnitude of the coefficients of tourism revenue remain unchanged. Similarly, the impact of most control variables changed slightly. These results support the robustness of this study. 

## 5. Discussion and Conclusions

This study highlights the importance of the relationship between tourism development and air pollution owing to its implications for sustainable tourism development. Considering the main criticisms of the tourism-induced EKC hypothesis, this study focuses on whether and how tourism can reduce or contribute to air pollution. Using a dataset of 283 prefecture-level cities in mainland China for 2005–2019, this study applies the PSTR approach to investigate the nonlinear effect of tourism development on air pollution proxied by PM_2.5_ emissions. Considering the internal heterogeneity across the different regions, we partitioned China into three distinct regions of East, central, and West cities, and further explored the validity of the tourism-induced EKC hypothesis as well as the impacts of other variables on air pollution at both the national and regional levels. 

Based on the theoretical framework of the EKC, our results show that the effects of tourism development on air pollution are neither linear nor nonlinear. Specifically, at the national level, the empirical results from the PSTR model reveal a significantly inverted U-shaped relationship between tourism and air pollution, supporting the validity of the conventional tourism-induced EKC hypothesis in China. When the number of tourists reaches the high regime (lnTA > 2.271), its effect on PM_2.5_ emissions changes from promoting to inhibiting. These findings are consistent with those of Bi and Zeng (2019) [29] and Zeng et al. (2021) [8], who found an inverted U-shaped impact of tourism on air pollution in China. However, our findings differ from those of other studies. For example, Zhang and Gao (2016) failed to provide evidence to support the tourism-induced EKC hypothesis in China [22]. Ahmad and Ma (2021) found that a 1% increase in tourism development can induce a 0.386% decrease in carbon emissions in Asian Tigers [12]. Saenz-de-Miera and Rosselló (2014) provided evidence that a 1% increase in tourist arrivals can induce a 0.45% increase in PM_10_ concentrations [14]. The differences among the above conclusions may be attributed to the differences in sample sizes, research methods, and sampled economies [12]. 

At the regional level, the results of the nonlinear marginal analysis indicate that the effects of tourism on PM_2.5_ emissions across different regions are heterogeneous. For the East cities, the effect of tourism on air pollution exhibits a U-shape, confirming the validity of the U-shaped tourism-induced EKC hypothesis. When the lnTA exceeds the second threshold value of 2.796, its effect on PM_2.5_ emissions transitions from inhibiting to promoting. For the Central and West cities, tourism exerts inhibitory effects on PM_2.5_ emissions in both the linear and nonlinear parts of tourist arrivals. These results imply that the tourism-induced EKC hypothesis does hold for central and western China. The findings differ slightly from those of Zhang and Gao (2016), who found that the tourism-induced EKC hypothesis is weakly supported in eastern and western China and fail to document the evidence in central China [22].

### 5.1. Theoretical and Policy Implications

Theoretically, our research contributes to the debate on the importance of the relationship between tourism development and air pollution. First, we propose a U–shaped tourism-induced EKC hypothesis which extends the theoretical connection between tourism development and air pollution under the traditional EKC framework. Our findings provide support for these two seemingly conflicting perspectives and empirically confirm that the nexus between tourism development and air pollution is nonlinear. This means that the mechanism through which tourism may have a significant impact on air pollution depends on different regimes of tourist arrivals. We have proved the nonlinear negatively shaped and inverted U-shaped relationships, which were automatically assumed in existing studies. We found empirical support for the extended tourism-induced EKC hypothesis that displays a U–shaped impact. 

Second, the nonlinear relationship between tourism and air pollution in our study may provide new insights into the ambiguous results of the linear model in the existing literature. Scholars should consider the nonlinear behavior of tourism-induced air pollution, not by including squared tourism specialization terms in their linear model, but by using a regime-switching model, such as the PSTR model. According to Lahouel et al. (2022) [49], the results in the linear models can be problematic, as they cannot portray all configurations with regard to nonlinear behavior. 

The findings of this study may provide valuable policy recommendations. First, the empirical results have documented that tourism can be responsible for air pollution reduction, implying that tourism is a cleaner industry than manufacturing and agricultural industries. However, as some scholars have suggested, tourism is an energy-intensive industry that produces a large amount of pollution; therefore, governments should not neglect the negative environmental externality of tourism and should formulate a series of stricter regulations and measures for the tourism industry to reduce pollution emissions and ensure the coordination between tourism and environmental protection. For example, the development of green tourism or eco-tourism can reduce air pollution. Second, the validity of the tourism-induced EKC is vital for considering sustainable growth and development for national and regional growth. At the national level, the early stages of tourism development should be of more concerns. Environment-friendly policies and measures should be issued by the government to maintain a balance between tourism development and environmental protection [10]. For the eastern region, we should focus more on the middle and later stages, as the substantial tourism development may lead to severe air pollution. In addition, the use of renewable energy and low-emission technology should be strongly encouraged by the government in the tourism sector to reduce air pollution.

### 5.2. Limitation and Future Research

Our study has some limitations. First, due to limited data availability, this study lacks discussions on the profiles of tourists (including local and international tourists) and other air pollutants such as CO, SO_2_, and NO_2_. According to Zhang et al. (2020), different tourism sectors have different effects on air pollution; thus, their profiles can influence this relationship [52]. Future research should focus on the effects of different types of tourism on different types of air pollution. Second, this study considered the level of tourist arrivals as the transition variable in the tourism-air pollution nexus. However, other relevant mechanisms through which tourism influences air pollution should be considered as transition variables in the PSTR model. Future studies should investigate the regime-switching effects of institutional quality, economic structure, and R&D investment on the relationship between tourism and air pollution. Finally, considering that air pollution may be relevant to the presence of old industries, heavy industries, and different industries in different regions to a large extent, future studies should compare the difference between the tourism air pollution nexus and the heavy industry air pollution nexus.

## Figures and Tables

**Figure 1 ijerph-19-08442-f001:**
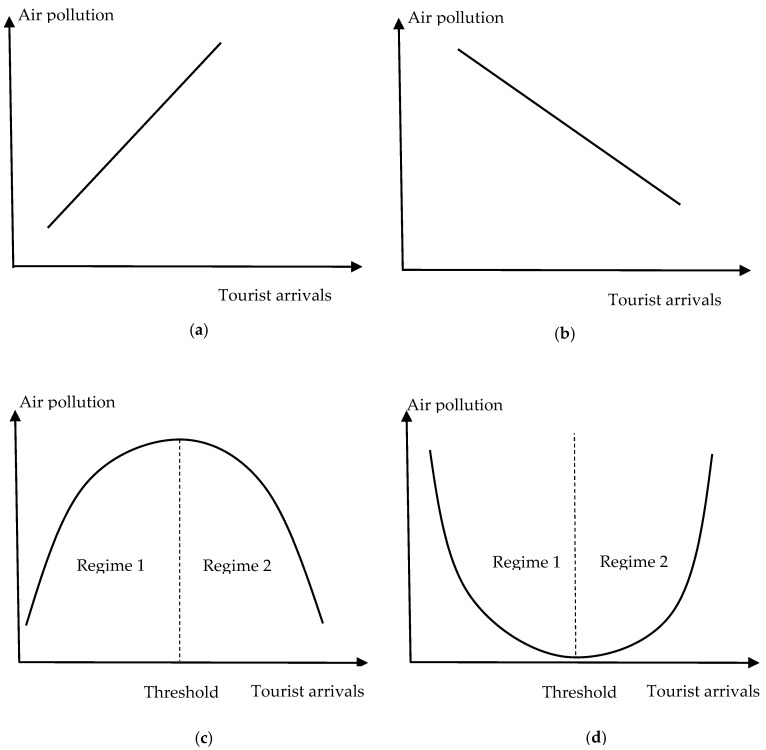
Theoretical connections between tourist arrivals and air pollution. (**a**) Positive; (**b**) negative; (**c**) the conventional tourism induced-EKC; (**d**) the U-shaped tourism induced-EKC.

**Figure 2 ijerph-19-08442-f002:**
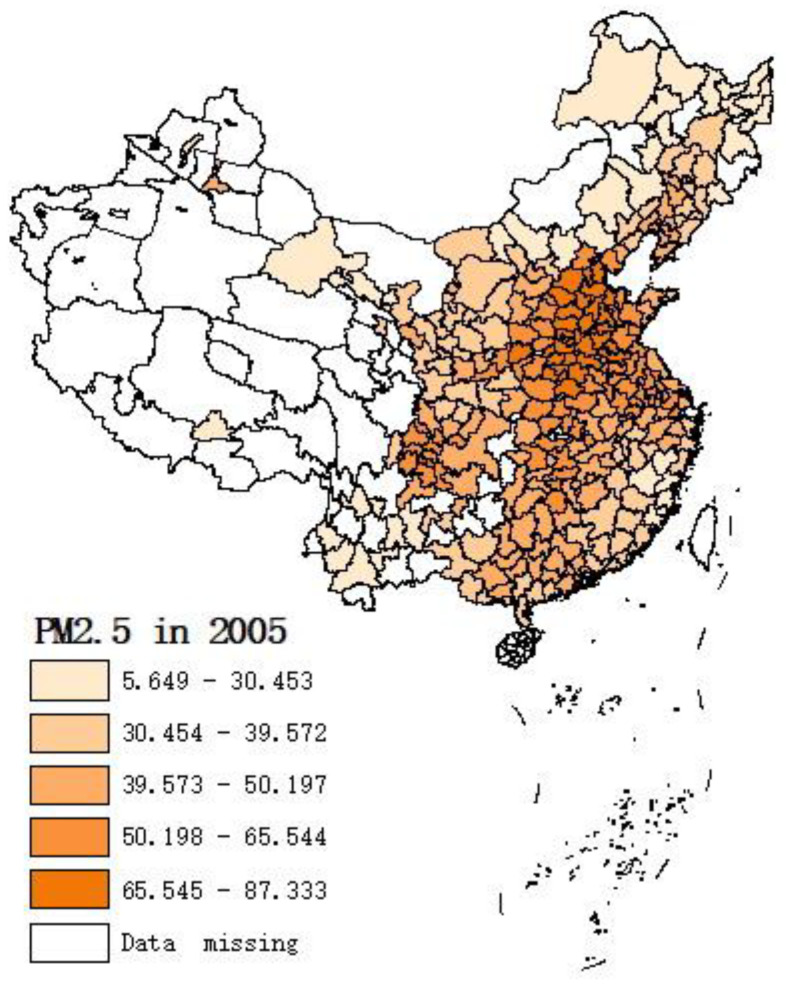
Spatial distribution of PM_2.5_ concentrations and tourist arrivals in 2005 and 2019.

**Figure 3 ijerph-19-08442-f003:**
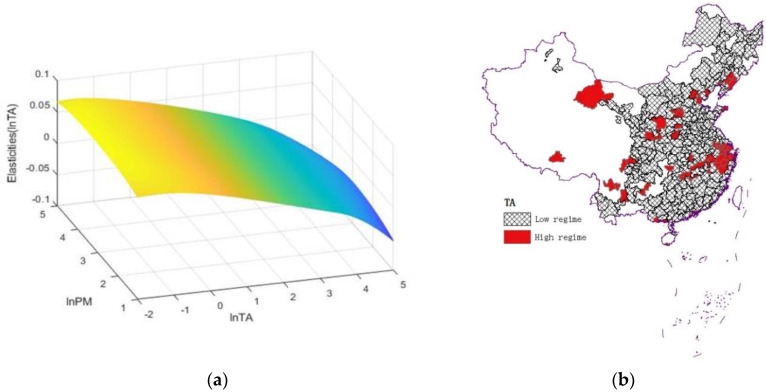
(**a**) The response surface of lnTA in relation to lnPM and lnTA. (**b**) The distribution of high-low tourism regimes.

**Figure 4 ijerph-19-08442-f004:**
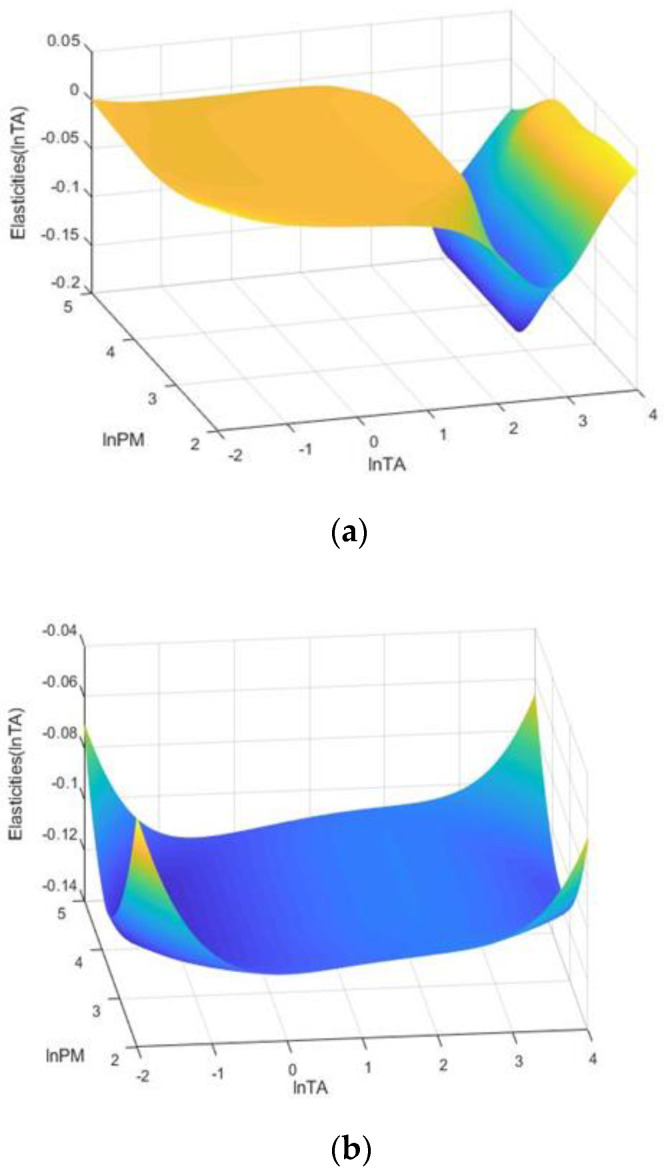
The 3D plot of the coefficient of lnTA in relation to the different level of lnTA. (**a**) East cities, (**b**) central cities, (**c**) West cities.

**Table 1 ijerph-19-08442-t001:** Descriptive statistics for the variables.

Variables	Definition	Obs	Mean	SD	Min	Max
lnPM	Logarithm of annual average PM_2.5_ concentrations (μg/m^3^)	4025	3.751	0.360	1.150	4.687
lnTA	Logarithm of the ratio of total tourist arrivals to the local inhabitants	4025	1.389	1.061	−1.986	4.286
lnPGDP	Logarithm of per capital GDP (CNY)	4025	2.333	0.105	0	2.752
lnDENS	Logarithm of the ratio of local inhabitants to urban area (million persons /10,000 km^2^)	4025	−1.185	0.933	−5.36	1.015
lnTECH	Logarithm of the ratio of Investment in Science and Technology to public expenditure of government	4025	−4.757	1.246	−15.538	0
lnINVEST	Logarithm of the ratio of fixed capital investment to GDP	4025	−0.410	0.601	−3.655	2.396
lnTRAFF	Logarithm of the ratio of the total passenger traffic volume of highway, railway transport, and civil aviation to local inhabitants	4025	2.577	0.882	−5.138	8.144
lnGREEN	Logarithm of the ratio of green coverage to urban area	4025	−0.989	0.397	−5.627	1.352

**Table 2 ijerph-19-08442-t002:** Results of the stationarity test.

	LLC	IPS	ADF−Fisher	ADF−PP	Conclusion
lnPM	1.162	14.464	178.191	400.978	Nonstationary
D.lnPM	18.425 ***	16.258 ***	1230.91 ***	3629.20 ***	Stationary
lnTA	9.376 ***	1.164	655.301 ***	1023.27 ***	Stationary
D.lnTA	11.049 ***	13.066 ***	1100.53 ***	2733.32 ***	Stationary
lnPGDP	−5.559 ***	9.169	393.726 ***	810.519 ***	Stationary
D.lnPGDP	−22.278 ***	−20.086 ***	1454.06 ***	4375.37 ***	Stationary
lnDENS	−15.329 ***	4.365	469.918	758.046 ***	Nonstationary
D.lnDENS	−29.112 ***	−12.645 ***	1011.48 ***	2763.52 ***	Stationary
lnINVEST	−7.382 ***	−5.387 ***	863.154 ***	1875.17 ***	Stationary
D.lnINVEST	−9.474 ***	−16.510 ***	1271.30 ***	3969.88 ***	Stationary
lnTECH	−13.546 ***	−3.489	850.017 ***	1181.34 ***	Stationary
D.lnTECH	−26.448 ***	−21.556 ***	1565.70 ***	2408.56 ***	Stationary
lnTRAFF	−4.067	6.397	372.726 ***	596.351 ***	Nonstationary
D.lnTRAFF	−13.763 ***	−7.725 ***	845.667 ***	2516.24 ***	Stationary
lnGREEN	−27.514 ***	−8.789 ***	898.184 ***	1378.12 ***	Stationary
D.lnGREEN	−23.663 ***	−20.052 ***	1447.31 ***	3458.67 ***	Stationary
Panel Kao residual cointegration test			−6.742 ***		

Notes: *** *p* < 0.01.

**Table 3 ijerph-19-08442-t003:** Linearity and remaining nonlinearity tests.

	H0: r = 0 vs. H1: r ≥ 1	H0: r = 1 vs. H1: r ≥ 2
LM	597.188 ***	25.860 **
LM_F_	92.497 ***	3.451
LRT	643.603 ***	25.939 **
	AIC	BIC
*m* = 1	−4.668	−4.644
*m* = 2	−4.667	−4.642

Notes: ** *p* < 0.05, *** *p* < 0.01.

**Table 4 ijerph-19-08442-t004:** Estimated results of the PSTR model for China.

	FE Model	PTR-FE Model	PSTR Model	
			Linear	Nonlinear
lnTA	−0.094 *** (0.003)		0.098 *** (0.010)	−0.155 *** (0.016)
lnPGDP	0.006 (0.022)	0.0005 (0.021)	0.288 *** (0.033)	−0.574 *** (0.053)
lnDENS	−0.128 *** (0.028)	−0.120 *** (0.027)	0.068 (0.046)	−0.389 *** (0.027)
lnTECH	0.030 *** (0.002)	0.029 *** (0.002)	0.006 (0.007)	0.062 *** (0.016)
lnINVEST	0.016 *** (0.004)	0.009 ** (0.004)	−0.014 (0.016)	0.043 (0.034)
lnTRAFF	0.045 *** (0.003)	0.041 *** (0.003)	0.032 *** (0.011)	0.003 (0.025)
lnGREEN	−0.0004 (0.006)	−0.003 (0.005)	−0.013 (0.020)	0.010 (0.057)
lnTA (lnTA < 2.045)		−0.067 * (0.003)		
lnTA (lnTA ≥ 2.045)		−0.097 *** (0.003)		
_cons	3.750 *** (0.062)	3.754 *** (0.061)		
Hausman test	81.58 ***			
γ			0.419	
c			2.294	
N		4245	4245	

Notes: standard error are in parentheses, * *p* < 0.1, ** *p* < 0.05, *** *p* < 0.01.

**Table 5 ijerph-19-08442-t005:** Estimated results of PSTR model for different regions.

Variables	East Cities			Central Cities		West Cities	
	Linear	Nonlinear		Linear	Nonlinear	Linear	Nonlinear
lnTA	−0.357 *** (0.077)	−0.028 *** (0.014)	0.344 *** (0.076)	−0.086 *** (0.025)	−0.044 (0.040)	−0.054 *** (0.015)	−0.087 *** (0.016)
lnPGDP	0.668 *** (0.090)	−0.533 *** (0.098)	−0.676 *** (0.100)	−0.013 (0.061)	0.069 (0.104)	0.056 (0.063)	−0.142 *** (0.034)
lnDENS	−0.019 (0.095)	−0.245 *** (0.052)	−0.019 *** (0.042)	−0.185 *** (0.059)	0.356 *** (0.041)	−0.017 (0.064)	−0.170 *** (0.014)
lnTECH	0.102 *** (0.028)	−0.010 (0.025)	−0.066 *** (0.022)	0.086 *** (0.016)	−0.098 *** (0.025)	0.016 *** (0.004)	0.018 * (0.010)
lnINVEST	−0.126 *** (0.052)	0.222 ** (0.054)	0.037 (0.038)	−0.359 *** (0.053)	0.608 *** (0.093)	0.039 *** (0.014)	−0.008 (0.019)
lnTRAFF	0.075 *** (0.047)	0.013 (0.051)	−0.042 (0.034)	0.128 *** (0.035)	−0.187 *** (0.057)	0.009 (0.007)	0.060 *** (0.014)
lnGREEN	0.650 (0.141)	−0.380 *** (0.142)	−0.561 *** (0.104)	−0.011 (0.072)	−0.019 (0.111)	0.014 * (0.008)	−0.049 * (0.028)
γ	−0.108; 6.463			0.213		1.500	
c	2.362; 2.796			1.893		1.633	
N	1500			1605		1140	

Notes: standard error are in parentheses, * *p* < 0.1, ** *p* < 0.05, *** *p* < 0.01.

**Table 6 ijerph-19-08442-t006:** Results of robustness check.

	Linear	Nonlinear
lnTR	0.012 ** (0.005)	−0.0003 (0.006)
lnPGDP	−0.003 (0.043)	−0.153 *** (0.019)
lnDENS	−0.162 ** (0.079)	−0.110 *** (0.009)
lnTECH	0.023 *** (0.003)	0.025 *** (0.006)
lnINVEST	−0.020 ** (0.008)	0.061 *** (0.011)
lnTRAFF	0.044 *** (0.005)	−0.016 * (0.008)
lnGREEN	0.007 (0.006)	−0.072 *** (0.017)
γ	2.195	
c	−1.853	
N	4245	

Notes: standard error are in parentheses, * *p* < 0.1, ** *p* < 0.05, *** *p* < 0.01.

## Data Availability

Not applicable.

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
