# Peer review of "Exploring the Effects of Tourism Development on Air Pollution: Evidence from the Panel Smooth Transition Regression Model"

_ijerph, 2022, doi:10.3390/ijerph19148442_

Round 1
Reviewer 1 Report
General comment:
On the one hand, the work has great potential, it presents well-conducted, scientific analyzes. On the other hand, the accuracy of the measurements and the input data are not fully discussed. There is also no in-depth cause-effect analysis. The same conclusions from other studies are repeated in different paragraphs, but it is difficult to find a specific numerical measure, for example, the percentage share of individual PM2.5 sources in the area. Authors are overusing inexact expressions enclosed in quotation marks. The idea of making a starting point for analyzes of what one group of "optimists" and the other group of "pessimists" believe in a scientific article in the technical field is a poorly understood choice for me. In such works, readers do not care about beliefs or not. You should write about research, numbers, and cause-effect relations.
plenty of editorial mistakes...
Abstract:
1. EKC is not introduced when 1st time used
2. Please use PM2.5 or PM2.5 instead of PM 2.5
Introduction:
1. Please also add information in $ beside local currency.
2. report of the Ministry of Culture and Tourism should be used as a reference
3. Why are you using once ' ' and in another sentence " "?4
4. Why in a scientific paper are you writing about beliefs? Optimists and pessimists believe... It is a technical, scientific paper. Please talk about numbers, research, and science. Not about beliefs.
5. OLS is not introduced when 1st time used
General comment to this section - what are primary PM2.5 sources in the investigated region in %? You are linking tourism with transportation etc. but what is the general distribution?
3.3 -> health effects of PM2.5 exposure should be moved to the introduction
What is the measurement accuracy? Did you try to compare it with reference stations?
Reviewer 2 Report
Good paper, methodologically sound. I would suggest improving the contextualization of the research.
I would suggest discussing the presence or absence of industries in the areas of study. Indeed, the concentration of heavy industries and other industries that impact rather negatively the air quality are not discussed. This obviously could be an area for further research. But it can also be in the present research a limitation and a methodological concern. Pollution from heavy industries, or the use of coal, could impact the results and be some kind of compounding factors.
Style throughout could be improved.
I have highlighted some points, but a thorough revision of the text is important.
In the file I have noted some additional points. The contribution is interesting but some caution with the dominating types of industries in the regions in China that are studied here is important.

Reviewer 3 Report
The paper provides a fundamental study into factors that may affect sustainable tourism, a topic of great interests. The authors presented data and associated models to analyze the underlining mechanism for PM2.5 generation, a yardstick the author focused in this study. The paper has been written clearly and concise, with minimum grammatical errors. However, there are some concerns associated with parts of the paper that the authors may need to pay special attention in revising this paper to an acceptable form.
1. Tourism being the focus of the study, the authors may wish to introduce more details about it. From what was gathered in the paper, the tourist arrivals seem to main indicator of tourism. Questions are, e.g., can there be components in terms of tourist composition, tourist activities,…etc? And will they be having different contribution to PM2.5 production? Can one identify the more critical ones? Will they contribute more significantly?
2. Based on the independent variables in the equations, it seems lacking factors depicting the proportion of tourism in the overall economies and their activity types and levels in the economy. For the factor representing transportation, it may be better to separate highway modes from public transportation modes, because their per capita contribution to PM2.5 production is usually less for the latter. It may not be ideal to lump them all together.
3. A few places in the paper seem to have texts or figures out of alignment that require the authors to double check and amend, e.g., page 5, 6 and 7, …etc, to name a few.
4. Some use of terms are not consistent, e.g., east cities, or eastern cities, etc. The authors may need to make them consistent.
5. The intervals in Figure 2 are very odd numbers. Are there any special reasons? It may be better if they’re defined in well-defined intervals.
6. In Table 2, “***” needs to be explained in a footnote like other tables that follow.
7. Page 3, section 3.3, fist sentence, “prefecture” should better be replaced with “county”, and for subsequent occurrences of such term.
8. Grammatical error, e.g., page 15, “we should more concern the early stage of tourism development”, should be written as, e.g., “The early stage of tourism development should be of more concerns”
9. Page 15, “environment friendly policies” should be “environmentally-friendly policies”.
10. Suggest the authors to carry out a more thorough review of the grammar and spelling of the writing in this paper.
Round 2
Reviewer 1 Report
Authors include my remarks in the revised version